# Amyloid-Targeting PET Tracer [^18^F]Flutemetamol Accumulates in Atherosclerotic Plaques

**DOI:** 10.3390/molecules24061072

**Published:** 2019-03-19

**Authors:** Sanna Hellberg, Johanna M.U. Silvola, Heidi Liljenbäck, Max Kiugel, Olli Eskola, Harri Hakovirta, Sohvi Hörkkö, Veronique Morisson-Iveson, Ella Hirani, Pekka Saukko, Seppo Ylä-Herttuala, Juhani Knuuti, Antti Saraste, Anne Roivainen

**Affiliations:** 1Turku PET Centre, University of Turku, FI-20520 Turku, Finland; sanna.hellberg@utu.fi (S.H.); johanna.silvola@gmail.com (J.M.U.S.); halilj@utu.fi (H.L.); max.kiugel@utu.fi (M.K.); olli.eskola@utu.fi (O.E.); juhani.knuuti@utu.fi (J.K.); antti.saraste@utu.fi (A.S.); 2Turku Center for Disease Modeling, University of Turku, FI-20520 Turku, Finland; 3Department of Vascular Surgery, Turku University Hospital, FI-20520 Turku, Finland; harri.hakovirta@tyks.fi; 4Medical Research Center and Nordlab Oulu, University Hospital and Research Unit of Biomedicine, Faculty of Medicine, University of Oulu, FI-90014 Oulu, Finland; sohvi.horkko@oulu.fi; 5GE Healthcare Ltd., Chalfont St Giles HP8 4SP, UK; Veronique.MorissonIveson@ge.com (V.M.-I.); Ella.Hirani@ge.com (E.H.); 6Department of Pathology and Forensic Medicine, University of Turku, FI-20520 Turku, Finland; psaukko@utu.fi; 7A.I. Virtanen Institute for Molecular Sciences, University of Eastern Finland, FI-70210 Kuopio, Finland; seppo.ylaherttuala@uef.fi; 8Turku PET Centre, Turku University Hospital, FI-20520 Turku, Finland; 9Heart Center, Turku University Hospital, FI-20520 Turku, Finland; 10Department of Clinical Medicine, University of Turku, FI-20520 Turku, Finland

**Keywords:** atherosclerosis, amyloid, imaging, positron emission tomography, autoradiography

## Abstract

Atherosclerosis is characterized by the accumulation of oxidized lipids in the artery wall, which triggers an inflammatory response. Oxidized low-density lipoprotein (ox-LDL) presents amyloid-like structural properties, and different amyloid species have recently been recognized in atherosclerotic plaques. Therefore, we studied the uptake of the amyloid imaging agent [^18^F]Flutemetamol in atherosclerotic plaques. The binding of [^18^F]Flutemetamol to human carotid artery plaque was studied in vitro. In vivo uptake of the tracer was studied in hypercholesterolemic IGF-II/LDLR^−/−^ApoB^100/100^ mice and C57BL/6N controls. Tracer biodistribution was studied in vivo with PET/CT, and ex vivo by gamma counter and digital ex vivo autoradiography. The presence of amyloid, ox-LDL, and macrophages in the plaques was examined by immunohistochemistry. [^18^F]Flutemetamol showed specific accumulation in human carotid plaque, especially in areas positive for amyloid beta. The aortas of IGF-II/LDLR^−/−^ApoB^100/100^ mice showed large thioflavin-S-positive atherosclerotic plaques containing ox-LDL and macrophages. Autoradiography revealed 1.7-fold higher uptake in the plaques than in a lesion-free vessel wall, but no difference in aortic tissue uptake between mouse strains were observed in the in vivo PET/CT. In conclusion, [^18^F]Flutemetamol binds to amyloid-positive areas in human atherosclerotic plaques. Further studies are warranted to clarify the uptake mechanisms, and the potential of the tracer for in vivo imaging of atherosclerosis in patients.

## 1. Introduction

Atherosclerosis is a chronic inflammatory disease characterized by the accumulation of lipids in vessel walls, especially oxidized low-density lipoproteins (ox-LDL), with these then triggering an inflammatory response [1]. While they first appear as distinct diseases, the pathologies of atherosclerosis and Alzheimer’s disease (AD) have similarities [2,3]. AD is a slowly developing disease characterized by the presence of extracellular aggregates of peptide fragments called amyloid beta (Aβ) in the brain. Aβ consists of 40 or 42 amino acid peptides, which are cleaved from amyloid precursor protein (APP). APP is predominantly expressed in neurons, but is also present as alternative splicing variants in platelets [4], in vascular endothelial cells [5], and in a soluble form in the circulation [6]. Recent studies show that higher concentrations of soluble APP and Aβ in the bloodstream are associated with higher cardiovascular risk [7,8], in addition to serum amyloid A [9]. Aβ is also present in atherosclerotic lesions [10,11,12]. It has been hypothesized that APP from platelets is further processed to Aβ in macrophages [11]. Since intraplaque hemorrhage is one of the hallmarks of rupture-prone plaques, it has been suggested that the presence of Aβ in an atherosclerotic plaque could be a sign of plaque’s vulnerability to rupture. Additionally, preclinical studies have shown links between APP and increased atherosclerosis. Transgenic APOE^−/−^ mice overexpressing APP show atherosclerotic lesions of increased size and with enhanced inflammation in comparison with APOE^−/−^ mice [13], while APOE^−/−^ mice lacking APP have less atherosclerosis than APOE^−/−^ mice [14]. Additionally, modification of LDL may induce amyloid-like structures that are recognized by the same receptors and stained by the same dyes as amyloid [15,16,17].

In AD, the positron emission tomography (PET) imaging of Aβ with the thioflavin-T analogue ^11^C-labeled Pittsburgh compound B (PIB) has been widely studied, as well as PET imaging with other tracers [18,19]. These tracers are derivatives of either benzothiazole (thioflavin-T analogues), stilbene or styrylpyridine (Figure 1), and they show distinct pharmacokinetic properties [19]. In a recent study, the Aβ-targeting PET tracer, stilbene derivative [^18^F]Florbetaben showed uptake in Aβ-containing advanced plaques in carotid arteries [20]. There is also a growing interest in the imaging of other types of amyloid beyond Aβ. For example, the imaging of cardiac amyloidosis with Aβ-targeting PET tracers has shown promising results [21,22]. Thus, Aβ-targeting tracers could potentially show uptake in other types of amyloid, in addition to Aβ.

[^18^F]Flutemetamol (2-(3-[^18^F]fluoro-4-(methylamino)phenyl)-1,3-benzothiazol-6-ol, [^18^F]3′F-PIB, is a novel fluorine-18 labeled analogue of [^11^C]PIB developed and marketed by GE Healthcare (Buckinghamshire, UK) [23]. The main advantages of [^18^F]Flutemetamol over [^11^C]PIB are the longer physical half-life of fluorine-18 facilitating the ex vivo autoradiography studies as well as lower positron range giving better spatial resolution in PET imaging of small targets. The aim of this study was to evaluate [^18^F]Flutemetamol in the imaging of atherosclerotic plaques. We studied the binding of [^18^F]Flutemetamol to human atherosclerotic plaque sections in vitro, while the in vivo and ex vivo biodistribution of [^18^F]Flutemetamol was studied in atherosclerotic and healthy mice, with more detailed uptake in the aorta being analyzed by digital autoradiography. The presence of macrophages, ox-LDL, and Aβ in murine atherosclerotic lesions was studied by immunohistochemistry.

## 2. Results

### 2.1. In Vitro Binding of [^18^F]Flutemetamol in Human Atherosclerotic Plaque

The human carotid artery sample showed partially calcified fibroatheroma-type plaque characteristics (Figure 2a). When incubated with [^18^F]Flutemetamol in vitro, the sections showed tracer accumulation in some of the non-calcified areas of the plaque (Figure 2b). [^18^F]Flutemetamol uptake was reduced when incubated with excess non-radioactive PIB, especially in the high-uptake areas (Figure 2c). The consecutive Aβ-stained sections showed positive staining in the area of the highest tracer uptake, whereas the calcified and fibrotic areas with low tracer uptake were negative for Aβ (Figure 2d–g).

### 2.2. Ex Vivo Autoradiography and Biodistribution in Mice

The atherosclerotic mice showed large fibroatheroma-type plaques in the aorta (Figure 3a), while no lesions were observed in the healthy controls. [^18^F]Flutemetamol accumulated in the aortas of both atherosclerotic and control mice (respective radioactivity concentrations: 0.81 ± 0.21 vs. 0.65 ± 0.20%ID/g, *p* = 0.12). In the autoradiography of atherosclerotic aortas (Figure 3b), the highest observed PSL/mm^2^ values were found in plaque calcifications (93 ± 30 PSL/mm^2^). By comparison, the mean uptake was 52 ± 8.6 PSL/mm^2^ in non-calcified plaques and 30 ± 5.0 PSL/mm^2^ in lesion-free vessel wall (*p* < 0.001). Uptake did not significantly differ between plaque areas of low and high macrophage infiltration (53 ± 13 vs. 52 ± 12 PSL/mm^2^). The adventitial uptake was 49 ± 8.7 PSL/mm^2^. The mice studied at the 60 min time point showed less uptake in the plaques than did the mice studied at 30 min post-injection (32 ± 4.6 PSL/mm^2^, *p* = 0.0039), whereas the uptake in the other ROIs was comparable. In the control mice, the largest radioactivity accumulation occurred in the adventitia (57 ± 10 PSL/mm^2^), while the uptake in healthy vessel wall was 32 ± 7.2 PSL/mm^2^.

The radioactivity concentration of [^18^F]Flutemetamol in selected tissues was measured at 30 min post-injection (Table 1). The highest [^18^F]Flutemetamol concentrations were observed in urine and kidneys (240 ± 180 and 16 ± 12 %ID/g, respectively, in atherosclerotic mice). The blood radioactivity concentration was 1.0 ± 0.58 %ID/g in atherosclerotic mice and 0.55 ± 0.21 %ID/g in control mice (*p* = 0.055). The uptake in the myocardium was low, at 0.49 ± 0.24 %ID/g in atherosclerotic mice and 0.37 ± 0.18 %ID/g in controls. The female mice tended to have higher %ID/g values in the tissues (Table 2). The results from the 60 min time point were not significantly different to the 30 min results (data not shown).

### 2.3. Histology and Immunohistochemistry

In histology studies, the mouse aortic sections showed cell-rich plaques with lipids and fibrotic tissue (Figure 4a,b). The cells were mainly macrophages, as can be seen by the Mac-3 staining (Figure 4c). The ox-LDL-immunostained sections showed that oxidatively modified LDL was abundant in the atherosclerotic lesions (Figure 4d). Thioflavin-S-stained sections showed green fluorescence under the microscope, both in the media and in the atherosclerotic lesions (Figure 4e). The observed fluorescence in the media was confirmed to be mainly autofluorescence (Figure 4f,g). Aβ immunostaining was the most prominent in the media, with only a minimal amount of positive staining being shown in the plaques (Figure 4h).

### 2.4. In Vivo PET/CT Imaging

The tracer was rapidly cleared from the blood circulation and accumulated in the gallbladder, urine, intestine, kidneys, and liver, representing the excretion routes (Figure 5). In other areas, the radioactivity concentration was low, generally around 0.5–1 %ID/mL. In almost all tissues, no significant differences in uptake were observed between the atherosclerotic and control mice (Table 1). The only exception was liver, where the atherosclerotic mice had lower uptake (4.8 ± 0.36 vs. 6.7 ± 0.65 %ID/mL, *p* = 0.032).

### 2.5. Metabolite Analysis

The mice rapidly metabolized [^18^F]Flutemetamol into various unidentified radioactive metabolites. At 30 min post-injection, intact [^18^F]Flutemetamol comprised 19% ± 5.4% and 17% ± 0.74% of total plasma ^18^F radioactivity in atherosclerotic and control mice, respectively. The vast majority of the ^18^F radioactivity in the urine was derived from metabolites, and only approximately 1% was derived from the intact tracer. An example of [^18^F]Flutemetamol high-performance liquid chromatography (HPLC) chromatogram is shown in Figure 6.

## 3. Discussion

In addition to AD, Aβ has been shown to be a relevant pathological feature in atherosclerosis, and potentially a marker of plaque vulnerability. Therefore, we aimed to detect this marker of vulnerable atherosclerotic plaques by [^18^F]Flutemetamol PET/CT. We showed specific focal binding of [^18^F]Flutemetamol in human atherosclerotic plaque, especially in areas positive for Aβ. We also showed preferential uptake by atherosclerotic plaques compared with lesion-free vessel wall in a mouse model, but the resolution of the PET was sub-optimal for in vivo visualization of lesions.

The proteins that form amyloid are diverse, and do not have similarities in their sequence or native structure, although they are all reactive to thioflavin. While the atherosclerotic plaques in this model were positive for thioflavin-S, they were negative for anti-Aβ immunohistochemistry. The observed thioflavin-S-positivity in the mouse atherosclerotic plaques may be driven by different types of thioflavin-reactive misfolded or aggregated proteins, such as ox-LDL. The presence of protein glycation-induced amyloid [24] in the mouse plaques in the current study is also possible, since the mouse model represents hyperglycemia in addition to atherosclerosis [25].

There is only a limited amount of data on the binding of Aβ targeting tracers to other types of amyloid. Both benzothiazole derivative [^11^C]PIB and styrylpyridine derivative [^18^F]Florbetapir have shown increased retention in the myocardium of patients with immunoglobulin light chain or transthyretin amyloidosis [21,22]. The Aβ targeting tracers have differences in their binding affinity, specificity, clearance, and metabolism. In addition to [^11^C]PIB, [^18^F]Flutemetamol has high specificity to Aβ and is metabolized to polar metabolites, which should not interfere in imaging lipid-rich atherosclerotic plaques [19]. The rapid metabolism in mice, however, might limit the bioavailability of the tracer. With non-Aβ targeting tracers, the PET imaging of ox-LDL in atherosclerotic plaques has been shown to be feasible in preclinical models [26,27].

Immunostaining showed the presence of Aβ in human atherosclerotic plaque; however, only a minimal amount of positive staining was observed in the mouse atherosclerotic plaques. This may be due to differences in plaque biology between the species, as mice rarely have intraplaque hemorrhage and therefore might not have platelet-derived Aβ in the plaques [8]. In humans, studies have shown variable results on the presence of Aβ in the vascular wall. Amyloid has been detected in vascular media in 97% of atherosclerotic patients over 50 years of age, whereas other studies found Aβ to be absent from vascular media. Amyloid has been detected in the atherosclerotic intima in 35–60% of patients [10,20,28], with the majority of Aβ deposition in human atherosclerotic plaques being shown to be located in macrophages [20]. The imaging of Aβ in mouse models of Alzheimer’s disease by in vivo PET imaging has proven to be challenging despite the presence of significant thioflavin positive lesions in the brain. This may be due to different amyloid plaque composition in mouse models compared to humans as well as lower density of high-affinity ligand binding sites [29,30]. Thus, despite the largely negative results of the imaging of the mouse atherosclerotic plaques with [^18^F]Flutemetamol, it may still show promise in patients with atherosclerosis.

The aortic radioactivity concentration was similar in atherosclerotic and healthy mice and was at the same level as the radioactivity of the blood pool. [^18^F]Flutemetamol has previously been shown to extensively bind with plasma proteins [31], which might explain the remaining blood radioactivity. Even elongating the time point to 60 min did not change the aorta-to-blood ratio. While we observed that the blood radioactivity concentration tended to be higher in atherosclerotic than control mice (1.0 ± 0.58 %ID/g vs. 0.55 ± 0.21 %ID/g) in the ex vivo measurement, no difference was detected in the in vivo PET data. This might be explained by the female mice in the ex vivo measurement, since they had significantly higher plasma radioactivity concentration than males (Table 2). The liver radioactivity in vivo was significantly lower in atherosclerotic mice than controls, and in the ex vivo measurement, the difference was also significant for male mice. This finding may reflect subtle impairment in the liver function in the atherosclerotic mouse model. However, it did not affect the metabolism of the tracer. The observed rapid radio-metabolism is in line with previous observations [31].

The current study offers preliminary data on the imaging of amyloid in atherosclerosis. The human plaque examined was from a single patient only, and more studies should be carried out to investigate the presence and correlation of amyloid with the severity of atherosclerosis. The mouse model atherosclerotic plaques did not have a significant amount of Aβ in them, which might explain the relatively low uptake. The in vivo imaging of the mouse plaques was limited by the small size of the plaques and the resolution of the PET camera. There was a discrepancy between the results in vitro on a patient sample and the results in vivo and ex vivo in mice. Conducting similar in vitro experiment on mouse aortic sections would have enabled more elaborate comparison of these data in mouse and man. Further mechanistic studies are needed to evaluate the binding target of [^18^F]Flutemetamol in atherosclerotic lesions.

## 4. Materials and Methods

### 4.1. Radiochemistry

[^18^F]Flutemetamol was synthesized according to methods described in the patent WO 2007/020400 A1 “Fluorination process of anilide derivatives and benzothiazole fluorinate derivatives as in vivo imaging agents”. FASTlab™ synthesizer (GE Healthcare, Waukesha, WI, USA) and single-use disposable cassettes designed for [^18^F]Flutemetamol production were used. The molar activity was 936 ± 390 GBq/μmol (mean ± SD, *n* = 11) at the end of synthesis, and the non-decay-corrected radiochemical yield was 20% ± 5%. Radiochemical purity was 95% ±1%.

### 4.2. In Vitro Binding in Human Atherosclerotic Plaque

To study the tracer specificity and binding in atherosclerotic plaques, [^18^F]Flutemetamol binding to human atherosclerotic plaque was investigated in vitro. The experiment was conducted as described previously [31], with small modifications. Human carotid artery plaque was obtained from endarterectomy surgery of a patient with a recent ischemic stroke. The patient study was conducted according to the Declaration of Helsinki, and the study protocol was approved by the ethics committee of Hospital District of Southwest Finland. The plaque sample was cut into 8 µm cryosections. The sections were thawed and pre-incubated in phosphate-buffered saline (PBS) for 20 min, followed by 30 min incubation in 15 mL PBS with or without the presence of 10 µM non-radioactive PIB (ABX Advanced Biomedical Compounds GmbH, Radeberg, Germany). Non-radioactive PIB was added in 15 µL of dimethyl sulfoxide (DMSO), and for the controls, a similar volume of DMSO was used. Then, [^18^F]Flutemetamol was added to final concentration of 0.1 MBq/mL and sections were incubated for another 30 min. Finally, the sections were washed with ice-cold PBS, rinsed with distilled water, and dried and opposed to an imaging plate (Fuji imaging plate BAS TR2025, Fuji Photo Film Co. Ltd., Tokyo, Japan) for 4 h.

### 4.3. Animals

Atherosclerotic and healthy mice were utilized to study the in vivo and ex vivo biodistribution of the tracer and its uptake in the aorta. A mouse model combining hypercholesterolemia and type 2 diabetes was used for this purpose. The mice are deficient in LDL receptor, synthesizing only apolipoprotein B100, while they overexpress insulin-like growth factor in pancreatic beta cells (IGF-II/LDLR^−/−^ApoB^100/100^) [25]. The mouse model closely resembles human familial hypercholesterolemia and has glucose intolerance and insulin resistance. The IGF-II/LDLR^−/−^ApoB^100/100^ mice were fed with a high-fat diet (0.2% total cholesterol, TD 88137, Envigo, Madison, WI, USA) for 4–5 months starting at the age of 2–3 months. Healthy C57BL/6N mice aged 7–11 months and fed with normal chow were used as controls. Mice were anesthetized with isoflurane during the experiments. The details of the animals in each experiment are described in Table 3. The study protocol was approved by the national Animal Experiment Board in Finland and the Regional State Administrative Agency for Southern Finland (Reference numbers ESLH-2009-06012/Ym-23 and ESAVI/1583/04.10.03/2012), and the studies were conducted in accordance with the European Union directive.

### 4.4. Ex Vivo Biodistribution and Aortic Autoradiography

For the ex vivo biodistribution analyses, ten IGF-II/LDLR^−/−^ApoB^100/100^ mice and eight C57BL/6N mice were studied at a time point 30 min after tracer injection. Three additional IGF-II/LDLR^−/−^ApoB^100/100^ mice were studied at a time point of 60 min. Isoflurane-anesthetized mice were injected with approximately 11 MBq of [^18^F]Flutemetamol via the tail vein. After the accumulation time, the mice were euthanized by cardiac puncture and cervical dislocation while under deep isoflurane anesthesia. The aorta was dissected from the ascending aorta to the level of the diaphragm and rinsed with saline. Radioactivity concentration in the aorta and other tissues was measured with a gamma counter (Triathler 3’’ Hidex, Turku, Finland) cross-calibrated with a dose calibrator (VDC-202, Veenstra Instruments, Joure, The Netherlands), and the tissues were weighed. Results were obtained as a percentage of the injected radioactivity dose per gram of tissue (%ID/g). 

The aortas of [^18^F]Flutemetamol-injected mice were prepared to serial longitudinal 20 and 8 µm cryosections. The sections were placed under an ionizing radiation-sensitive imaging plate (Fuji imaging plate BAS TR2025, Fuji Photo Film Co. Ltd., Tokyo, Japan) for a minimum of 4 h followed by scanning with a phosphoimager (Image analyzer BAS-5000, Fuji Photo Film Co. Ltd., Tokyo, Japan) for digital autoradiography. The autoradiography data were corrected for injected radioactivity dose/mouse weight and the radioactivity decay during exposure. Data were analyzed with TINA 2.1 software (Raytest isotopemessgeräte GmbH, Straubenhardt, Germany), and the results were obtained as photo-stimulated luminescence per square millimeter (PSL/mm^2^) with the background subtracted. Images of hematoxylin-eosin-stained 20 µm sections were carefully co-registered with the autoradiographs. Regions of interest (ROIs) were defined on (1) non-calcified plaque excluding media, (2) healthy vessel wall, (3) adventitia, and (4) calcification. The validity of the analysis method was previously reported [32].

### 4.5. Immunohistochemistry

The human carotid artery plaque sections utilized for autoradiography were stained with hematoxylin and eosin. Consecutive sections of the same plaque were stained with anti-Aβ antibody (Novocastra Mouse Monoclonal NCL-B-Amyloid, Leica Biosystems Newcastle Ltd., Newcastle upon Tyne, UK), with Vectastain Elite Avidin-Biotin Complex (ABC) kit (Vector Laboratories Inc., Burlingame, CA, USA) reagents being utilized in the staining protocol where applicable. Sections were fixed with formalin and pre-treated with formic acid followed by endogenous peroxidase blocking with normal goat serum. Primary antibody (1:50 dilution) was incubated overnight at +4 °C, followed by secondary antibody (biotinylated anti-mouse IgG 1:200 dilution). Vectastain Elite ABC reagent and peroxidase substrate 3,3′-diaminobenzidine (DAB) were utilized for detection, and sections were counterstained with hematoxylin.

The 20 µm mouse aortic cryosections were stained with hematoxylin and eosin. Consecutive 8 µm sections were stained with Movat’s pentachrome, anti-Mac-3, anti-ox-LDL antibody, or thioflavin-S dye. Macrophage immunostaining was performed with an anti-Mac-3 antibody (BD Pharmingen, Franklin Lakes, NJ, USA; Clone M3/84, 1:5000 dilution), as described previously [32]. Anti-ox-LDL antibody (mouse IgM, against malondialdehyde-modified LDL, Clone 5.HMC+10_101) [33] was kindly provided by Sohvi Hörkkö. Sections were fixed with acetone, and endogenous peroxidase was blocked with H_2_O_2_. After subsequent blocking with bovine serum albumin, sections were incubated with biotinylated primary antibody (1:100 dilution), followed by Vectastain Elite ABC reagent (Vector Laboratories Inc., Burlingame, CA, USA) and peroxidase substrate DAB. Finally, sections were counterstained with hematoxylin. Thioflavin-S staining was performed to detect amyloid in the plaques. After fixation in formalin, sections were incubated with 1% thioflavin-S (T1892, Sigma-Aldrich, St. Louis, MO, USA) and washed with 80% ethanol and water. Sections were imaged with a fluorescence microscope (AxioVert 220M, Zeiss, Jena, Germany) immediately after staining, and again a week later to obtain images of the autofluorescence.

Additional mouse aortic sections were stained with anti-Aβ antibody (AB2539, 1:1000 dilution, Abcam, Cambridge, UK) to show the presence and localization of Aβ in the aortas of the atherosclerotic mice. The staining protocol was similar to the protocol utilized in the staining of carotid artery plaques, except that the sections were pre-treated with boiling citrate buffer and the secondary antibody used was biotinylated anti-rabbit IgG.

### 4.6. In Vivo PET/CT Imaging

Three atherosclerotic and three control mice (Table 1) were imaged on a small animal PET/CT scanner (Inveon Multimodality, Siemens Medical Solutions, Knoxville, TN, USA). They were injected with approximately 5 MBq of [^18^F]Flutemetamol via the tail vein, and a dynamic 40 min PET scan was started from the time of injection. After PET imaging, 150–200 µL of contrast agent (eXIA 160XL, Binitio Biomedical Inc., Ottawa, ON, Canada) was injected and high-resolution CT was performed. The CT acquisition consisted of 271 projections with an exposure time of 1250 ms, x-ray voltage of 80 kVp, and an anode current of 500 µA for a full 360° rotation.

The dynamic PET data were reconstructed with a 2-dimensional ordered-subsets expectation maximization algorithm to time frames of 30 × 10 s, 15 × 60 s, and 2 × 600 s. The reconstructed images had a matrix size of 128 × 128 × 159, and pixel size of 0.776 × 0.776 × 0.796 mm, and attenuation correction was performed with the CT modality. CT images were reconstructed with a filtered back-projection algorithm (pixel size, 0.094 × 0.094 × 0.094 mm). The PET data were analyzed with Carimas 2.8 software (Turku PET Centre, Turku, Finland). In brief, volumes of interest (VOIs) were defined in various tissues on the basis of the CT anatomy, and their time-activity curves were extracted. The VOI for blood was located in the vena cava. The 30 min time point was used in the analysis of tissue uptake.

### 4.7. Radiometabolite Analysis

The in vivo metabolism of [^18^F]Flutemetamol was determined using a HPLC method. Three IGF-II/LDLR^−/−^ApoB^100/100^ and three healthy C57BL/6N mice were used for these analyses (Table 1). Approximately 10 MBq of [^18^F]Flutemetamol was injected via the tail vein. Blood was collected at 30 min after injection by cardiac puncture under deep isoflurane anesthesia. Urine was collected from the bladder at the same time point. Plasma was separated, plasma and urine proteins were precipitated, and the samples were analyzed using HPLC with a μBondapak C18 column (125 Å, 10 μm, 7.8 × 300 mm; Waters, Milford, MA, USA) and an acetonitrile/50 mmol/L phosphoric acid gradient with a flow rate of 6.0 mL/min.

### 4.8. Statistical Analysis

All values are expressed as mean ± standard deviation. *p* values less than 0.05 are regarded as statistically significant. Student’s *t*-test was used to compare differences between atherosclerotic and control mice. ANOVA with a Tukey HSD or a Tamhane correction was used for multiple comparisons. The Tamhane correction was used when variances were observed to be unequal in a Levene’s test; otherwise, the Tukey HSD correction was used.

## 5. Conclusions

The results of this study support previous findings on the use of amyloid-targeting tracers for imaging of atherosclerotic plaque vulnerability. Further mechanistic studies are warranted to properly characterize the binding of [^18^F]Flutemetamol to different types of amyloid and ox-LDL.

## Figures and Tables

**Figure 1 molecules-24-01072-f001:**
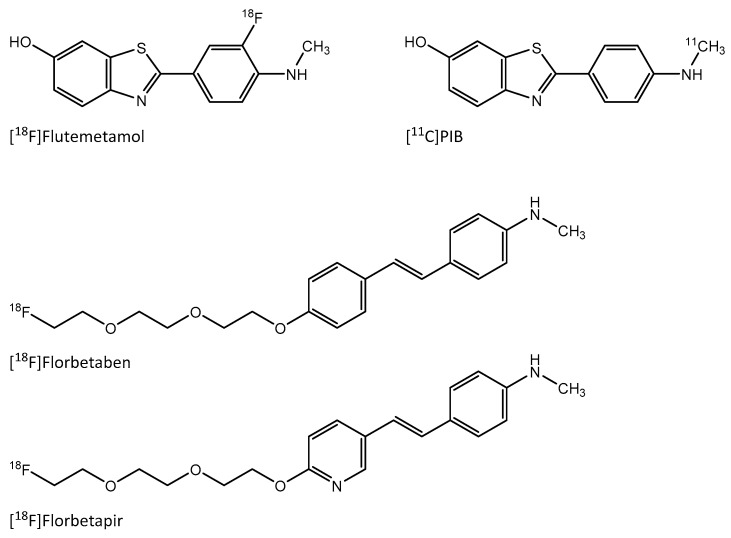
PET tracers targeting amyloid beta. [^11^C]PIB and [^18^F]Flutemetamol are benzothiazole derivatives based on the thioflavin-T molecule. [^18^F]Florbetaben is a derivative of stilbene and [^18^F]Florbetapir a derivative of styrylpyridine.

**Figure 2 molecules-24-01072-f002:**
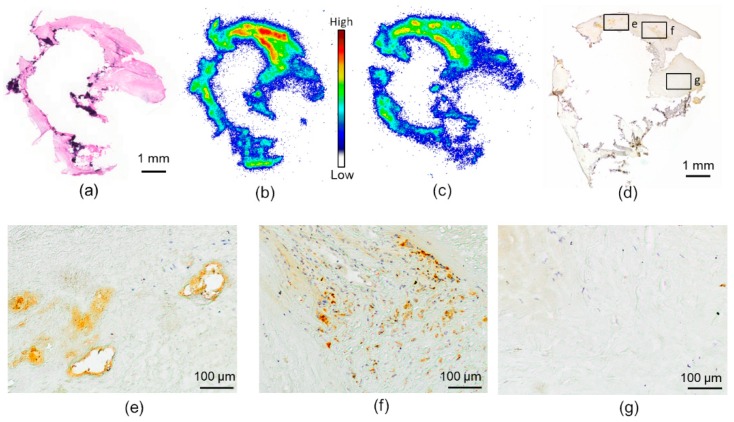
[^18^F]Flutemetamol in vitro autoradiography of human carotid atherosclerotic plaque. (**a**) Hematoxylin-eosin-stained section of human carotid endarterectomy sample. Calcified areas are seen as dark purple; the rest of the plaque represents fibroatheroma. (**b**) [^18^F]Flutemetamol in vitro autoradiography of the same section as in A. High uptake is seen as red in specific areas of the plaque. (**c**) In vitro autoradiography with an excess amount of unlabeled Pittsburgh compound B (PIB) shows decreased accumulation of [^18^F]Flutemetamol in the plaque. The same scale as in (**b**). (**d**) Aβ staining of a consecutive section of the carotid artery plaque. (**e**–**g**) Magnification of the areas specified in (**d**). Aβ-positivity is observed in the areas with high [^18^F]Flutemetamol accumulation, whereas no staining is seen in the area of low accumulation.

**Figure 3 molecules-24-01072-f003:**
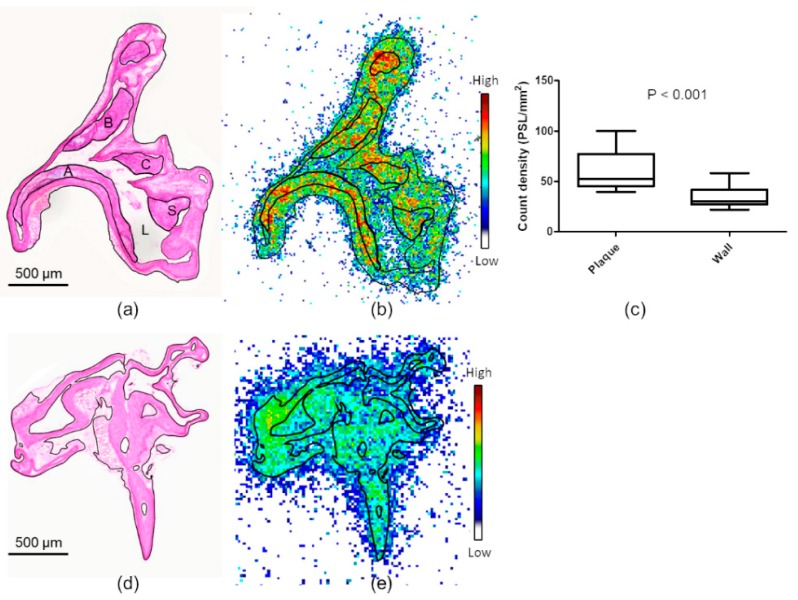
[^18^F]Flutemetamol ex vivo autoradiography of mouse aortic sections. (**a**) Hematoxylin-eosin-stained section of longitudinally cut atherosclerotic mouse aorta shows large fibroatheroma-type plaques. The plaques in aortic arch (A), brachiocephalic trunk (B), left common carotid (C) and subclavian artery (S) are circled in the figure. The lumen is annotated with L. (**b**) Corresponding [^18^F]Flutemetamol ex vivo autoradiography. (**c**) Quantitative results of the autoradiography in atherosclerotic mice. Uptake in the plaques is significantly higher than the uptake in apparently normal vessel wall. (**d**) Hematoxylin-eosin stained aortic section of a healthy mouse. (**e**) Corresponding [^18^F]Flutemetamol ex vivo autoradiography. Note that the autoradiographs in (**b**,**e**) were acquired separately and are not corrected for radioactivity decay during exposure, and thus the scales are not comparable.

**Figure 4 molecules-24-01072-f004:**
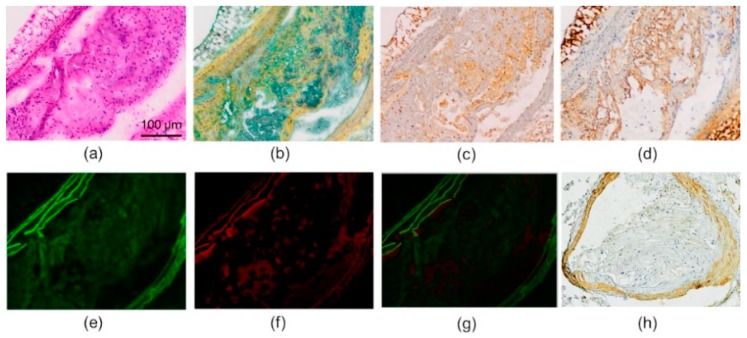
Histology of the mouse atherosclerotic plaques. (**a**) Hematoxylin and eosin staining shows cell-rich plaque in the brachiocephalic artery. (**b**) Movat’s pentachrome staining shows fibrotic tissue as yellow and lipids as green. (**c**) Mac-3 immunostaining of the plaque shows abundant infiltration of macrophages. (**d**) Ox-LDL immunostaining. (**e**) Fluorescent thioflavin-S amyloid staining. (**f**) The same section as in E imaged after the fading of thioflavin-S fluorescence shows the areas of autofluorescence. (**g**) A merged image of E and F localizes the autofluorescence to mainly the media and areas with only low thioflavin-S fluorescence. (**h**) Amyloid beta immunostaining of another brachiocephalic plaque shows intense staining in vessel wall, but only minimal staining in the plaque intima.

**Figure 5 molecules-24-01072-f005:**
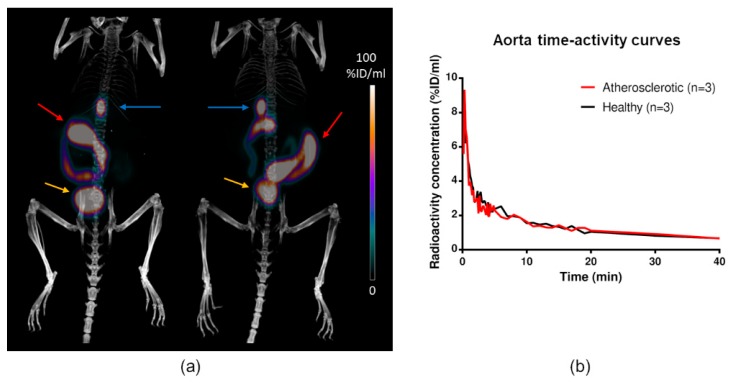
Examples of [^18^F]Flutemetamol in vivo PET/CT images. (**a**) PET/CT of a healthy C57BL/6N mouse (left) and atherosclerotic IGF-II/LDLR^−/−^ApoB^100/100^ mouse (right) at 30 min post-injection. High radioactivity is seen in the gallbladder (blue arrows), intestine (red arrows), and urinary bladder (yellow arrows). The kidneys are also faintly visible. (**b**) Time-activity curves in aorta show no difference between the atherosclerotic and healthy mice.

**Figure 6 molecules-24-01072-f006:**
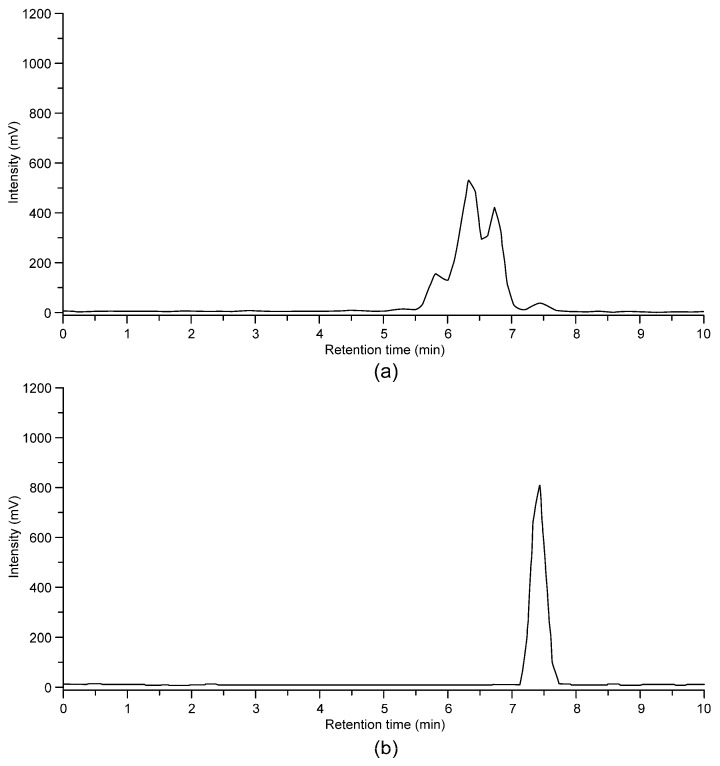
Radiometabolism of [^18^F]Flutemetamol evaluated with HPLC. (**a**) A chromatogram of C57BL/6N mouse plasma at 30 min post-injection shows several unidentified radioactive metabolites. Only a small fraction of the tracer is intact. (**b**) [^18^F]Flutemetamol standard.

**Table 1 molecules-24-01072-t001:** The in vivo and ex vivo biodistribution of [^18^F]Flutemetamol in mice at 30 min post-injection.

	In Vivo PET (%ID/mL)			Ex Vivo Biodistribution (%ID/g)		
Tissue	Atherosclerotic (*n* = 3)	Control (*n* = 3)	*p* Value	Atherosclerotic (*n* = 10)	Control (*n* = 8)	*p* Value
Aorta	0.87 ± 0.091	0.82 ± 0.10	0.55	0.81 ± 0.21	0.65 ± 0.20	0.12
Blood	1.0 ± 0.27	1.5 ± 0.043	0.15	1.0 ± 0.58	0.55 ± 0.21	0.055
Brain	1.2 ± 0.038	1.4 ± 0.36	0.55	N/A	N/A	N/A
Gallbladder	110 ± 32	97 ± 22	0.64	N/A	N/A	N/A
Intestine	78 ± 48	84 ± 19	0.89	N/A	N/A	N/A
Kidneys	9.8 ± 2.0	11 ± 3.7	0.62	16 ± 12	14 ± 4.4	0.65
Liver	4.8 ± 0.36	6.7 ± 0.65	0.032	4.6 ± 2.1	8.0 ± 5.2	0.074
Muscle	0.30 ± 0.032	0.32 ± 0.026	0.52	0.26 ± 0.092	0.29 ± 0.10	0.44
Myocardium	0.82 ± 0.0021	1.0 ± 0.12	0.13	0.49 ± 0.24	0.37 ± 0.18	0.25
Urine	69 ± 26	100 ± 28	0.30	240 ± 180	120 ± 65	0.081
WAT	0.56 ± 0.37	0.49 ± 0.22	0.83	0.48 ± 0.20	0.40 ± 0.32	0.51

Data are expressed as percentage of injected radioactivity per milliliter or gram of tissue (mean ± SD). WAT = white adipose tissue. *p* value derived from Student’s *t*-test, atherosclerotic vs. control.

**Table 2 molecules-24-01072-t002:** Ex vivo biodistribution results calculated separately for male and female mice.

Tissue	Control (Male, *n* = 8)	Atherosclerotic (Male *n* = 5)	*p* Value (vs. Control)	Atherosclerotic (Female, *n* = 5)	*p* Value (vs. Control)	*p* Value (vs. Male Atherosclerotic)
Aorta	0.65 ± 0.20	0.70 ± 0.11	0.60	0.92 ± 0.20	0.063	0.11
Blood	0.55 ± 0.21	0.59 ± 0.031	0.65	1.3 ± 0.53	0.045	0.055
Kidney	14 ± 4.4	9.1 ± 3.4	0.064	23 ± 12	0.21	0.084
Liver	8.0 ± 5.2	3.3 ± 0.88	0.038	5.8 ± 2.0	0.32	0.048
Muscle	0.29 ± 0.10	0.19 ± 0.023	0.024	0.32 ± 0.079	0.60	0.014
Myocardium	0.37 ± 0.18	0.33 ± 0.054	0.58	0.66 ± 0.21	0.057	0.036
Plasma	0.79 ± 0.24	1.0 ± 0.22	0.14	1.8 ± 0.82	0.11	0.10
Urine	120 ± 65	280 ± 180	0.15	190 ± 130	0.42	0.46
WAT	0.40 ± 0.32	0.41 ± 0.090	0.89	0.55 ± 0.23	0.38	0.32

Data are expressed as percentage of injected radioactivity dose per gram of tissue (mean ± SD). WAT = white adipose tissue.

**Table 3 molecules-24-01072-t003:** Details of the studied animals.

	Biodistribution + ARG		In Vivo PET/CT		Metabolite Analysis	
	Atherosclerotic	Control	Atherosclerotic	Control	Atherosclerotic	Control
No. of animals (m/f)	10 (5/5)	(8/0)	3 (3/0)	3 (3/0)	3 (3/0)	3 (3/0)
Age (months)	6–8	7–11	8	3	9	7
High-fat diet (months)	4–5	N/A	4	N/A	5	N/A
Weight (g)	35 ± 7.0	44 ± 3.3	42 ± 3.4	33 ± 2.7	38 ± 6.3	37 ± 8.3
Injected radioactivity (MBq)	11 ± 0.95	11 ± 2.1	4.7 ± 1.1	5.4 ± 1.3	9.2 ± 1.3	11 ± 0.44

ARG = autoradiography. Weight is expressed as mean ± SD. For biodistribution and autoradiography at 60 min post-injection, an additional three female mice were studied (characteristics similar to those in the biodistribution + ARG group).

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
