# Peer review of "Amyloid-Targeting PET Tracer [18F]Flutemetamol Accumulates in Atherosclerotic Plaques"

_molecules, 2019, doi:10.3390/molecules24061072_

Round 1

Reviewer 1 Report

The authors aimed to investigate the uptake of an amyloid-targeting PET probe in both human and mouse model atherosclerotic plaques. The seemingly specific In vitro binding of [18F]Flutemetamol to human plaques justifies the approach and the development of animal models for more a more extensive in vivo investigation. Thus, the research question is relevant and well laid out. The results of their in vivo investigation open up further avenues for exploration particularly where the development and validation of small animal models for this line of research is concerned.

General Comments and Criticisms:

Regarding structure and content:

             Introduction

The introduction is well constructed, however, it would benefit from a brief discussion of the chemical and biological differences (if any) between [11C]PiB, [18F]Florbetaben, and [18F]Flutemetamol (Including a figure of the chemical structures). What is the advantage/novelty in using [18F]Flutemetamol in this study over the previously investigated compounds such as [11C]PiB? Is it related to the half-live of the isotope (11C Vs 18F)?

                Material and Methods 

i)              The methods section would also benefit from a more specific description of the “in vitro binding in human atherosclerotic plaques” protocol. What amount (volume) of non-radioactive PiB was used. How much [18F]Flutemetamol (volume or MBq amount) and what buffer was used. This information would be vital to anyone planning to repeat this experiment or compare this data to their own.

                Results:

i)               The authors examine the in vitro binding of [18F]Flutemetamol to ex vivo sections of human carotid atherosclerotic plaques and attempt to block specific binding using excess non-radioactive PiB. Was a titration performed to quantify the extent of the blocking effect (ie 10 µmol vs 100 µmol vs 1000 µmol). This would be helpful in gauging the binding specificity of the tracer.

ii)              Figure 2 would benefit substantially from the inclusion of histology/autoradiography images of a control mouse although this is not completely necessary.

iii)            Table 1: would be more easily understood as a figure (Eg. Bar Chart)

                Discussion

i)               There is a large amount of overlap between the introduction and discussion, particularly in the first 2 paragraphs of the discussion. This could be improved upon.

ii)              A key point in the manuscript is the binding specificity of the tracer to Aβ vs other amyloid types. This is important when considering the exact amyloid make-up of the investigated plaques and how that effects the uptake of a tracer that targets . It would be interesting to see if blocking with non-radioactive PiB has an effect on the [18F]Flutemetamol binding in the mouse model (specifically the ex vivo autoradiography experiments), thus giving an indication of whether [18F]Flutemetamol binding specificity is comparable across the 2 autoradiography experiments (human vs animal). To this end an in vitro autoradiography experiment on mouse aorta would make these two data sets more comparable.

Specific suggestions regarding language and formatting:

Errors regarding the correct naming of radiopharmaceuticals are made throughout the manuscript. Eg, 18F-Flutemetamol should be written as [18F]Flutametamol. 11C-PIB, should be written as [11C]PiB. The names of these radiopharmacuticals should be adjusted throughout the manuscript to align with “Consensus nomenclature rules for radiopharmaceutical chemistry” (https://www.sciencedirect.com/science/article/pii/S0969805117303189?via%3Dihub)

Line 48: Although they first appear as…

Line 49-51: Consider revising sentence structure. “the brain” is the object and should be at the end of the sentence.

Line 53: Consider grammar “differently splicing variants in platelets”

Line 129: In Histology studies…

Author Response

Replies to Reviewer #1

“The authors aimed to investigate the uptake of an amyloid-targeting PET probe in both human and mouse model atherosclerotic plaques. The seemingly specific In vitro binding of [18F]Flutemetamol to human plaques justifies the approach and the development of animal models for more a more extensive in vivo investigation. Thus, the research question is relevant and well laid out. The results of their in vivo investigation open up further avenues for exploration particularly where the development and validation of small animal Aβ models for this line of research is concerned.

General Comments and Criticisms:

Regarding structure and content:

Introduction

The introduction is well constructed, however, it would benefit from a brief discussion of the chemical and biological differences (if any) between [11C]PiB, [18F]Florbetaben, and [18F]Flutemetamol (Including a figure of the chemical structures). What is the advantage/novelty in using [18F]Flutemetamol in this study over the previously investigated compounds such as [11C]PiB? Is it related to the half-live of the isotope (11C Vs 18F)?”

Authors’response:

We thank the reviewer for this consideration. We agree that this is an important topic to cover in the Introduction. We have added a new figure of the chemical structures (Figure 1) and discussed the differences of the tracers shortly in the Discussion. In the Introduction we shortly present the advantages of [18F]Flutemetamol over [11C]PIB in preclinical setting:

“The main advantages of [18F]Flutemetamol over [11C]PIB are the longer physical half-life of fluorine-18 facilitating the ex vivo autoradiography studies as well as lower positron range giving better spatial resolution in PET imaging of small targets.”

“Material and Methods 

i)    The methods section would also benefit from a more specific description of the “in vitro binding in human atherosclerotic plaques” protocol. What amount (volume) of non-radioactive PiB was used. How much [18F]Flutemetamol (volume or MBq amount) and what buffer was used. This information would be vital to anyone planning to repeat this experiment or compare this data to their own. “

Authors’response:

Thank you for the comment. The protocol description has now been revised to be more precise. We now write:

The sections were thawed and pre-incubated in phosphate-buffered saline (PBS) for 20 minutes, followed by 30 minute incubation in 15 ml PBS with or without the presence of 10 µM non-radioactive PIB (ABX Advanced Biomedical Compounds GmbH, Radeberg, Germany). Non-radioactive PIB was added in 15 µl of dimethyl sulfoxide (DMSO) and for the controls, similar volume of DMSO was used. Then, [18F]Flutemetamol was added to final concentration of 0.1 MBq/ml and sections were incubated for another 30 minutes. Finally, the sections were washed with ice-cold PBS, rinsed with distilled water, and dried and opposed to an imaging plate (Fuji imaging plate BAS TR2025, Fuji Photo Film Co. Ltd, Tokyo, Japan) for 4 hours.”

“Results:

i)    The authors examine the in vitro binding of [18F]Flutemetamol to ex vivo sections of human carotid atherosclerotic plaques and attempt to block specific binding using excess non-radioactive PiB. Was a titration performed to quantify the extent of the blocking effect (ie 10 µmol vs 100 µmol vs 1000 µmol). This would be helpful in gauging the binding specificity of the tracer.”

Authors’ response:

We agree that this kind of titration experiment would have been useful in evaluating the blocking effect. Unfortunately, such titration experiment was not performed in this study. The use of 10 µM concentration for the blocking has been thoroughly optimized earlier, and therefore we decided to use only that.

ii)   “Figure 2 would benefit substantially from the inclusion of histology/autoradiography images of a control mouse although this is not completely necessary.”

Authors’ response:

We thank for this comment. The autoradiography and histology images of a healthy mouse have been added to the figure (Figure 3 of the revised manuscript).

iii) “Table 1: would be more easily understood as a figure (Eg. Bar Chart)”

Authors’ response:

Thank you for this comment. We would like to keep it as a table, because the graph would have almost 40 bars in total and would be difficult to read.

“Discussion

i)    There is a large amount of overlap between the introduction and discussion, particularly in the first 2 paragraphs of the discussion. This could be improved upon.”

Authors’ Response:

We agree that there was significant overlap. These sections have been revised.

ii)   “A key point in the manuscript is the binding specificity of the tracer to Aβ vs other amyloid types. This is important when considering the exact amyloid make-up of the investigated plaques and how that effects the uptake of a tracer that targets Aβ. It would be interesting to see if blocking with non-radioactive PiB has an effect on the [18F]Flutemetamol binding in the mouse model (specifically the ex vivo autoradiography experiments), thus giving an indication of whether [18F]Flutemetamol binding specificity is comparable across the 2 autoradiography experiments (human vs animal). To this end an in vitro autoradiography experiment on mouse aorta would make these two data sets more comparable.”

Authors’ response:

We thank the Reviewer for this suggestion and agree that such an experiment would have been interesting and could have made it easier to compare the mouse and human data. Unfortunately this was technically challenging and we could not add this kind of experiment to the manuscript in this short revision time.

“Specific suggestions regarding language and formatting: 

Errors regarding the correct naming of radiopharmaceuticals are made throughout the manuscript. Eg, 18F-Flutemetamol should be written as [18F]Flutametamol. 11C-PIB, should be written as [11C]PiB. The names of these radiopharmacuticals should be adjusted throughout the manuscript to align with “Consensus nomenclature rules for radiopharmaceutical chemistry” (https://www.sciencedirect.com/science/article/pii/S0969805117303189?via%3Dihub)”

“Line 48: Although they first appear as…

Line 49-51: Consider revising sentence structure. “the brain” is the object and should be at the end of the sentence.

Line 53: Consider grammar “differently splicing variants in platelets”

Line 129: In Histology studies…”

Authors’ Response:

Thank you for pointing out errors in nomenclature and language. The radiopharmaceutical names have now been changed to be in agreement with the nomenclature consensus. The indicated sentences have also been revised.

Reviewer 2 Report

The authors examine the characteristics of 18F-flutemetamol binding to Aβ and modified lipids, and to evaluate 18F-flutemetamol in the imaging of atherosclerotic plaques. The following comments need to be addressed before publication.

1)    The radiosynthesis of 18F-flutemetamol should be briefly described. Its radiochemical yields should be provided. It is unclear how 18F-flutemetamol was synthesized by hand or using an automated system?  

2)    18F-flutemetamol in vitro autoradiography of human carotid in Figure 1b and c needs to be under the same scale bar.

3)    The time activity curve of aortas from both IGF-II/LDLR-/-ApoB100/100 and healthy C57BL/6N mice should be presented in a Figure.

4)    HPLC chromatogram of radiometabolite analysis at differnet time points should be presented in a Figure.

5)    Representative 18F-flutemetamol in vivo PET/CT image in IGF-II/LDLR-/-ApoB100/100 mice should also be added to Figure 4.

Author Response

Replies to Reviewer #2

“The authors examine the characteristics of 18F-flutemetamol binding to Aβ and modified lipids, and to evaluate 18F-flutemetamol in the imaging of atherosclerotic plaques. The following comments need to be addressed before publication.

1)    The radiosynthesis of 18F-flutemetamol should be briefly described. Its radiochemical yields should be provided. It is unclear how 18F-flutemetamol was synthesized by hand or using an automated system? “ 

Authors’ Response:

Thank you for this suggestion. We have followed the advice and now write:

“[18F]Flutemetamol was synthesized according to methods described in the patent WO 2007/020400 A1 “Fluorination process of anilide derivatives and benzothiazole fluorinate derivatives as in vivo imaging agents”. FASTlab™ synthesizer (GE Healthcare, Waukesha, USA) and single-use disposable cassettes designed for [18F]Flutemetamol production were used. The molar activity was 936 ± 390 GBq/μmol (mean ± SD, n = 11) at the end of synthesis, and the non-decay-corrected radiochemical yield was 20% ± 5 %. Radiochemical purity was 95% ± 1%.”

“2)    18F-flutemetamol in vitro autoradiography of human carotid in Figure 1b and c needs to be under the same scale bar.”

Authors’ response:

This is an important point. These autoradiographs are indeed in the same scale, which has now been added to the Figure 2 in the revised manuscript and further clarified in the figure legend.

“3)    The time activity curve of aortas from both IGF-II/LDLR-/-ApoB100/100 and healthy C57BL/6N mice should be presented in a Figure.”

Authors’ response:

We thank for this comment. The time-activity curves have now been added to the Figure 5 of the revised manuscript.

“4)    HPLC chromatogram of radiometabolite analysis at differnet time points should be presented in a Figure.”

Authors’ response:

Thank you for this suggestion, we agree. The HPLC chromatogram of mouse plasma sample has now been added (Figure 6 of the revised manuscript).

“5)    Representative 18F-flutemetamol in vivo PET/CT image in IGF-II/LDLR-/-ApoB100/100 mice should also be added to Figure 4. “

Authors’ response:

We thank for this comment. The PET/CT of both mouse strains have now been included in the Figure 5 of the revised manuscript.

Reviewer 3 Report

The manuscript submitted by Hellberg et al. reports the evaluation of F-18 labeled Flutemetamol for imaging atherosclerotic plaque using a transgenic mouse model. Uptake of F-18 labeled Flutemetamol was conducted via in vivo PET/CT imaging and ex vivo biodistribution studies. The study design was thorough and detailed experimental procedures were provided. Despite negative results, potential causes for the low signal from mouse atherosclerotic plaque were provided and well discussed. Therefore, this manuscript could be accepted for publication after the following suggested changes have been made:

·                 Figure 4: the scale of the color bar should be provided.

·                 Table 1: the atherosclerotic group of mice (n = 10) used for ex vivo biodistribution study should be divided into female (n = 5) and male (n = 5) groups, and their corresponding P values should be calculated separated. This is because the tracer might have different pharmacokinetics in male and female mice.

·                 It is well documented that mouse amyloid plaque has less C-11 PIB binding sites than human amyloid plaque. Could this be a factor that low signal from mouse atherosclerotic plaque was observed in this report?

Author Response

Replies to Reviewer #3

“Comments and Suggestions for Authors

The manuscript submitted by Hellberg et al. reports the evaluation of F-18 labeled Flutemetamol for imaging atherosclerotic plaque using a transgenic mouse model. Uptake of F-18 labeled Flutemetamol was conducted via in vivo PET/CT imaging and ex vivo biodistribution studies. The study design was thorough and detailed experimental procedures were provided. Despite negative results, potential causes for the low signal from mouse atherosclerotic plaque were provided and well discussed. Therefore, this manuscript could be accepted for publication after the following suggested changes have been made:

Figure 4: the scale of the color bar should be provided.”

Authors’ response:

We thank for pointing this out. The whole figure has been revised per other reviewer’s suggestion (Figure 5 in the revised manuscript). The scale has also been added.

“Table 1: the atherosclerotic group of mice (n = 10) used for ex vivo biodistribution study should be divided into female (n = 5) and male (n = 5) groups, and their corresponding P values should be calculated separated. This is because the tracer might have different pharmacokinetics in male and female mice.”

Authors’ response:

We thank for pointing this out. We have now calculated the biodistribution results separately for male and female mice and observed subtle differences. The comparison between female and male atherosclerotic mice separately to controls has been added (Table 2 of the revised manuscript). The results are also discussed in the Discussion section. 

“It is well documented that mouse amyloid plaque has less C-11 PIB binding sites than human amyloid plaque. Could this be a factor that low signal from mouse atherosclerotic plaque was observed in this report?”

Authors’ response:

We thank the Reviewer for this comment. We agree that this may contribute to the result. We discussed this further in the manuscript. We now write:

“The imaging of Aβ in mouse models of Alzheimer’s disease by in vivo PET imaging has proven to be challenging despite the presence of significant thioflavin positive lesions in brain. This may be due to different amyloid plaque composition in mouse models compared to humans as well as lower density of high-affinity ligand binding sites [29,30].”

Round 2

Reviewer 1 Report

The authors have significantly improved the manuscript and I thank them for their time in considering my suggestions. 

The manuscript is acceptable in its current form. However, Figure 1 is not of the highest quality and this could be improved by using ChemDraw and the "ACS Document 1996 stationary." However, this is a very minor concern and I would leave it to the editors to decide if this figure is of acceptable quality for publication.

Author Response

Reviewer 1:

Comments and Suggestions for Authors

The authors have significantly improved the manuscript and I thank them for their time in considering my suggestions. The manuscript is acceptable in its current form. However, Figure 1 is not of the highest quality and this could be improved by using ChemDraw and the "ACS Document 1996 stationary." However, this is a very minor concern and I would leave it to the editors to decide if this figure is of acceptable quality for publication.

Authors’ reply: We thank the reviewer for this positive response. We agree that the Figure 1 quality was not optimal, and thus it was re-done in ChemDraw to obtain better quality.